# Bot fit: A novel approach to assessing lower limb muscular strength

Byungmun Kang[1], DaeEun Kim[1]*, Changmin Lee[1], Dongwoo Kim[2], Hwang-Jae Lee[2], Dokwan Lee[2], YoonMyung Kim[3], Hyung-Gyu Jeon[4], Kyunghwan Jung[5]

1 Biological Cybernetics Lab, School of Electrical and Electronic Engineering, Yonsei University, Seoul, Republic of Korea, 2 Bot Fit T/F, New Business Team, Samsung Electronics, Suwon, Republic of Korea, 3 University College, Yonsei University International Campus, Incheon, Republic of Korea, 4 Sports Rehabilitation Lab, Department of Physical Education, Yonsei University, Seoul, Republic of Korea, 5 Department of Sports, Leisure, and Recreation, Soonchunhyang University, Chungnam, Republic of Korea

* daeeun@yonsei.ac.kr

## Abstract

This study introduces Bot Fit, an innovative hip-joint exoskeleton designed for the objective assessment of lower limb muscular strength and exercise performance. A total of 25 participants underwent baseline strength assessments using conventional fitness equipment before performing resistance exercises under two controlled conditions: a 10-second test without speed restrictions and a 30-second test at a constant speed. Bot Fit recorded motor signals and performance metrics, capturing key indicators such as repetitions and movement patterns. To analyze the relationship between muscle parameters and exercise performance, we employed correlation analysis, hierarchical cluster analysis (HCA), and regression analysis. The results demonstrated strong associations between lower limb strength and key exercises, including squats, knee-ups, and reverse lunges, particularly in short-duration and constant-speed conditions. HCA successfully clustered participants based on three exercise-related metrics with an accuracy of 0.89, while the regression model achieved a correlation coefficient of 0.86, indicating high statistical power. These findings suggest that Bot Fit provides a reliable, data-driven method for muscular strength assessment, offering valuable insights for optimizing exercise programs, preventing injuries, and enhancing physical performance. The integration of wearable robotic technology into health and fitness monitoring enables personalized exercise interventions and supports evidence-based decision-making in sports science, rehabilitation, and performance enhancement.

## Introduction

Muscle strength assessment plays a crucial role in evaluating overall health and well-being. It is essential for determining health status [1], designing effective exercise and rehabilitation programs [2], and preventing or managing age-related

**Data availability statement:** Data Availability Statement Anonymized datasets sufficient to replicate the analyses and findings reported in the manuscript are provided as Supporting Information: (1) muscle_param.csv, containing muscle strength parameters used in the regression analysis; (2) demographics.csv, containing participant age, height, weight, and BMI; (3) angle_data.zip, including raw joint angle data (left and right legs) from all exercise protocols; and (4) kneeup_correction.csv, containing processed knee-up exercise data underlying Figure 6. Researchers with justified purposes may contact the corresponding author for general inquiries about the study. However, all data requests must be directed to a non-author data contact: Minchul Kim, Samsung Electronics Co., Ltd. Email: min1chul.kim@samsung.com This contact will review and handle data requests in accordance with Samsung Electronics' confidentiality and intellectual property policies.

**Funding:** The authors declare that financial support was received for the research, authorship, and/or publication of this article. This study was supported by Samsung Electronics, Republic of Korea. Additional support was provided by the National Research Foundation of Korea (NRF), funded by the Korean government (MSIT) (Grant No. 2020R1A2B5B01002395), and by the Korea Institute for Advancement of Technology (KIAT), funded by the Korean government (MOTIE) (Grant No. P0020535, The Competency Development Program for Industry Specialists). The funders had no role in study design, data collection and analysis, decision to publish, or preparation of the manuscript.

**Competing interests:** The authors have declared that no competing interests exist.

conditions [3]. Given its significance, various methods have been developed to measure muscular strength across different contexts, including health assessment, exercise planning, and performance evaluation in both athletic and rehabilitative settings [4,5].

Traditional approaches, such as the One-Repetition Maximum (1RM) test for lower limb strength [6] and Handheld Dynamometry [7,8], as well as bodyweight exercises like push-ups and squats [9,10], are widely used. However, these methods often have limitations, including reliance on specialized equipment and environmental constraints. As an alternative, gym-based resistance training has been recognized as an effective means to enhance both muscular strength and overall health [11].

In addition to these conventional methods, bodyweight exercises offer a practical approach to strength assessment without the need for specialized equipment. One widely used method involves analyzing exercise performance or tracking workout records within a defined timeframe. To evaluate the effectiveness of different training protocols, studies frequently compare pre- and post-training performance in athletes, utilizing metrics such as squat jumps and countermovement jumps to assess muscular strength [9,12].

Wearable robots present a promising alternative by providing adaptive resistance that overcomes the limitations of traditional strength assessment methods. These devices can be integrated into various activities, such as walking, working, and exercising [13,14], dynamically adjusting assistance based on user movements to enhance support and adaptability.

Lee et al. pioneered the development of an intelligent hip-assist exoskeleton robot, now recognized as Bot Fit, which operates in both assistance and resistance modes [15,16]. Studies have demonstrated that incorporating this robot into single-session resistance exercise programs enhances physical performance [17,18]. Given the critical role of lower-limb musculature in human locomotion and overall strength, this study aims to estimate and assess lower-limb strength using exoskeleton robots.

Lower-limb exoskeletons have also been widely employed for motor recovery in patients with neurological injuries [19]. Additionally, research has shown that joint mechanics in the lower limbs vary with gait and incline: uphill walking shifts positive power from the ankle to the hip, whereas running primarily relies on ankle mechanics. These findings offer valuable insights into the design of exoskeletons that optimize energy injection, extraction, and transfer [20]. Another study demonstrated that quadratic regression models of sagittal kinematic and kinetic gait parameters during very slow walking (0.2–0.8 m/s) correlate strongly with speed, without altering gait strategy. This allows for more precise scaling of lower-limb exoskeleton trajectories for individuals with paralysis [21]. Collectively, the literature provides a comprehensive review of advancements in exoskeleton technology, particularly in lower-limb applications [22,23].

Most research on exoskeleton robotics has focused on rehabilitation for individuals with movement impairments [24]. For example, motion intent recognition using surface electromyography (sEMG) signals and support vector machines (SVM) has enabled real-time gait switching in exoskeletons, significantly improving human-machine interaction and rehabilitation safety [25]. While numerous studies have

demonstrated the effectiveness of exoskeleton-assisted exercises, developed techniques for quantifying muscle force impairment, and enhanced upper-limb function assessments in virtual reality-based stroke rehabilitation [17,18,26], a critical gap remains in the systematic evaluation of muscular strength in healthy individuals using exoskeleton robots.

Traditional muscle strength assessment methods often lack clear individual performance indicators and require additional sensors. In contrast, our approach leverages an exoskeleton robot to directly quantify muscle strength while simultaneously capturing user movements, ensuring high reproducibility and reliability during resistance exercises.

In our study, the hip-joint exoskeleton robot Bot Fit was used to facilitate controlled resistance exercises for 25 healthy adults performing five various resistance exercises. By analyzing motor signals, we developed performance metrics that, when correlated with baseline muscle strength, reliably reflected the effectiveness of each exercise (Fig 1). This method extends the application of exoskeleton robots beyond rehabilitation, offering a practical and equipment-free solution for muscle strength assessment that eliminates the need for fixed measurement locations.

## Methods

### Experimental platform

**Hip-joint exoskeleton.** The hip-joint exoskeleton Bot Fit, a wearable robot developed by Samsung Electronics Co., Ltd., Korea, delivers resistance torque to the hip joints during exercise. It is a lightweight (2.9 kg), slim, comfortable, and

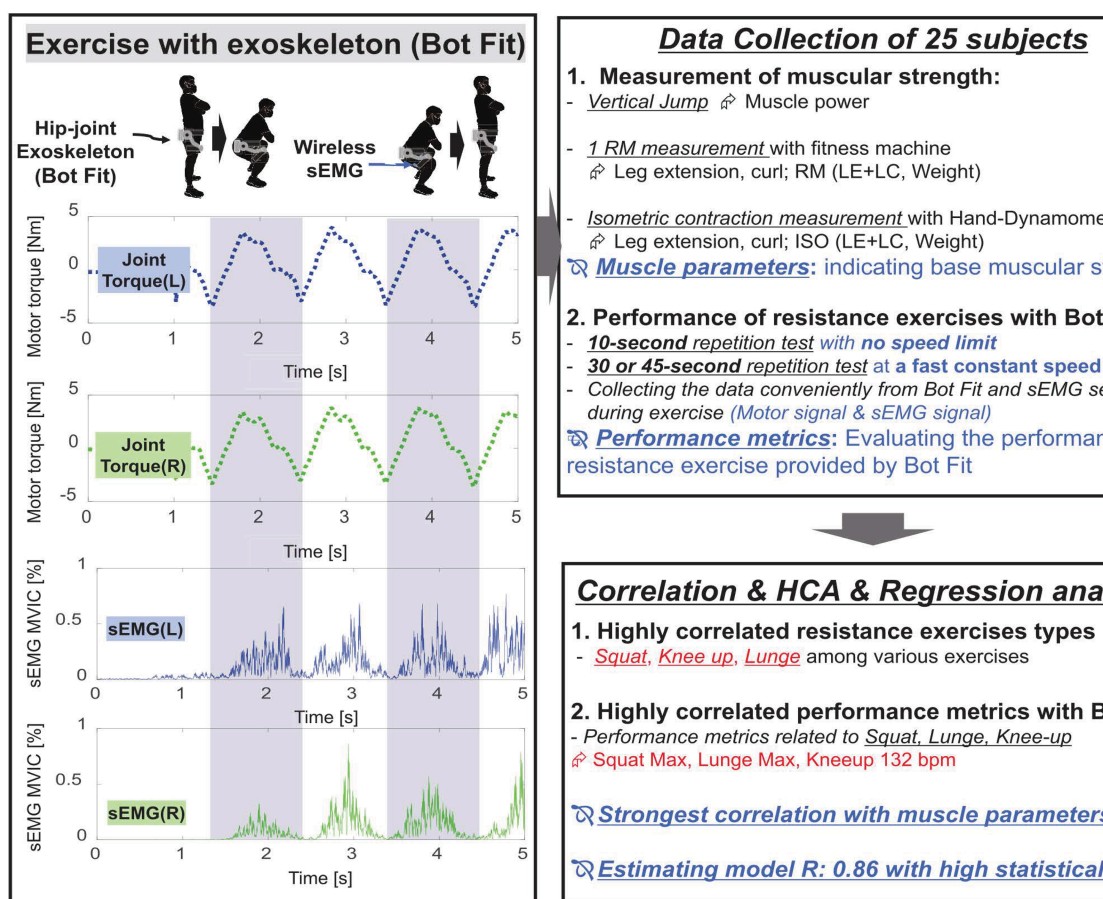

**Fig 1**. **Overview of the model of estimating muscle strength.**

active exercise assistance robot. Bot Fit consists of a pair of actuators that generate resistance to the left and right hip joints, a fabric waist belt and adjustable waist support, and a couple of thigh frames that transmit resistance torque from the actuator to the legs.

Two BLDC motors (Brushless DC Motors), known for their high rotational speeds and precise control enabling smooth speed adjustments, are mounted near the hip joint to generate auxiliary or resistance torque [27]. The generated torque is transmitted to each joint via a multi-stage gear system [17,18,28]. The inertial measurement unit (IMU) sensor on the back side of the robot monitors the subject's activity, estimates walking or exercising speed, and determines whether the subject starts or stops walking.

The torque in exercise is crafted to create resistance in a specific direction, depending on the type of exercise. The torque of the Bot Fit's motors is contingent on the joint angle, typically reaching a maximum of 10Nm. Sensors measure the direction and angle of movement during exercise, generating torque in the opposite direction to provide resistance to the user's movement. Moreover, a feedback loop monitors the rhythm of the exercise motion to generate the appropriate torque corresponding to the exercise speed.

**Wireless sEMG sensors.** We utilized wireless sEMG sensors (Delsys Trigno System, Boston, MA, USA) to measure thigh muscular activity. These sensors were affixed to four different regions on each thigh: rectus femoris, vastus lateralis, biceps femoris, and semitendinosus of each leg. The wireless sEMG sensor, a commonly used dry electrode-type device (5 × 1 mm), is composed of 99.9% silver. It is applied without causing any inconvenience to the user and can function for up to 8 hours when fully charged.

These muscles' activity was assessed by attaching the sEMG sensors to the four major thigh muscles, and this process was integrated into the existing muscular strength measurement procedure using exercise equipment and the Bot Fit.

### Subjects and experimental protocol

**Subjects.** A total of 25 adults (17 males, 8 females) were enrolled in this study. All participants received detailed information about the study procedures and provided written informed consent prior to participation. The study protocol was reviewed and approved by the Institutional Review Board (IRB) of Yonsei University (IRB No. 7001988-202204-HR-1538-02).

Recruitment commenced following IRB approval on 06/04/2022. The initial experiment began on 01/06/2022, and supplementary experiments were conducted between 01/07/2022 and 09/08/2022. Participant recruitment concluded on 31/12/2022. All research personnel completed certified online ethics training before conducting the experiments. Data collection, processing, and finalization were completed on 06/04/2023, and the study completion report was submitted to the IRB on 08/05/2023.

The subject selection criteria were as follows: 1) Healthy adults in their 20s to 30s without a surgical history related to internal medical conditions, such as cardiovascular diseases, musculoskeletal disorders, and neurological disorders; 2) Individuals willing to participate voluntarily, without any reluctance to wear wearable robots and capable of attaching wireless sEMG sensors for measuring muscle activity; 3) Those who willingly participated in the study.

Moreover, the exclusion criteria for subjects encompassed individuals meeting the following conditions: 1) severe communication impairments resulting from cognitive disorders or aphasia; 2) Severe internal medical conditions, such as cardiovascular diseases and conditions affecting lower limb muscles; 3) Individuals with a BMI level of 40 or above (considering subject safety due to the inappropriate size of the exoskeleton robot used in the study); 4) Any other cases where the researcher deemed participation in the study as inappropriate.

Therefore, the subjects in this paper fulfilled the mentioned selection criteria and can be broadly characterized as healthy individuals involved in light exercise approximately 1 to 2 times a week. Detailed information about the subjects is provided in Table 1.

**Table 1**. **Characteristics of the subjects.** SD standard deviation.

| Characteristic | Values |
|---|---|
| Sex (male/female) | 17/8 |
| Age (mean± SD) | 27.12 ± 3.66 [years] |
| Weight (mean± SD) | 69.63 ± 13.03 [kg] |
| Height (mean± SD) | 169.45 ± 8.53 [cm] |
| BMI (mean± SD) | 24.25 ± 6.1 [$kg/m^2$] |

**Experimental protocols.** The experimental protocols consisted of two measurement phases for each subject: a pre-measurement phase and an exercise measurement phase utilizing the maximum resistance mode of Bot Fit (Fig 2).

**Pre-measurement phase** During the pre-measurement phase, we assessed the baseline muscular strength of the subjects using fitness equipment without Bot Fit. Muscle parameters reflecting each subject's baseline strength were recorded (Table 2). Muscular power was evaluated through the Vertical Jump (VJ) test, where the best record from three trials was selected. This test requires participants to bend their knees in place and jump vertically [29].

Muscular strength, represented by the one-repetition maximum (1RM), was measured using leg extension (LE) and leg curl (LC) fitness machines (Life Fitness, Franklin Park, Illinois, USA). In the leg extension exercise, participants lifted the maximum possible weight by extending their knee and foot while seated, primarily engaging the rectus femoris [6,30]. In contrast, the leg curl exercise, performed in a lying position, required participants to lift the maximum weight by flexing the knee and foot, primarily activating the biceps femoris [31].

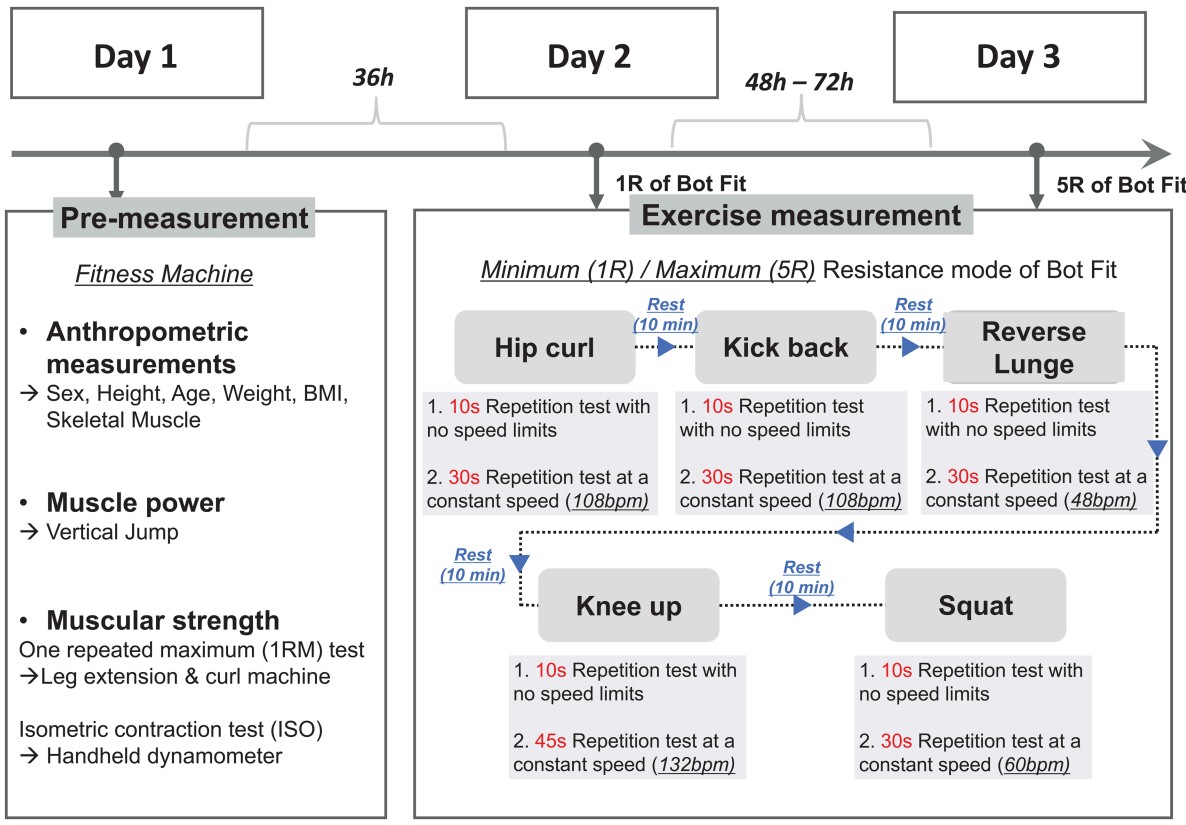

**Fig 2**. **Design of experimental protocols.**

**Table 2. Description of muscle parameters.** These parameters obtained from the pre-measurement indicate the base muscular strength of each subject.

| Parameters | Description |
|---|---|
| VJ | Max Height of three trials of vertical jump. (p-value: 0.348) |
| RM (LE+LC, Weight) | Normalization values of sum of the measured values by leg extension (LE) and leg curl (LC) with fitness machine to the subject's weight. (p-value: 0.337) |
| ISO (LE+LC, Weight) | Normalization values of sum of the measured values by leg extension (LE) and leg curl (LC) with isometric contraction to the subject's weight. (p-value: 0.565) |
| Total performance | Comprehensive muscular strength index; calculated as the sum of z-scores for VJ, RM (LE+LC, Weight) and ISO (LE+LC, Weight) [38]. (p-value: 0.812) |

Additionally, lower limb muscle strength was assessed using a hand-held dynamometer (EasyForce, GMT Ltd, Bury Saint Edmunds, UK). The strength of the rectus femoris and biceps femoris was measured by securing the dynamometer to the ankle while subjects remained in a seated position. They were instructed to exert maximum force for five seconds with their knee maintained at a 90-degree angle [32,33].

To prevent injuries and reduce strain between exercises and before proceeding to the next exercise protocol, subjects were given adequate stretching and rest periods. The pre-measurement phase was conducted during the first visit on Day 1, and to minimize any potential influence on subsequent experiments, the exercise measurement phase was scheduled 1–2 days later.

**Exercise measurement phase.** During the exercise measurement phase, subjects performed five resistance exercises using Bot Fit on Day 2 and Day 3, as shown in Fig 2. These exercises were chosen for their compatibility with Bot Fit, emphasizing pelvic motion control through bilateral leg movements in the sagittal plane. Bot Fit applies resistance only in this plane, preventing lateral pelvic movement.

The selected exercises engaged muscles where sEMG sensors were attached, including the rectus femoris, vastus lateralis, biceps femoris, and semitendinosus [34–36]. Subjects performed exercises under two conditions. The first condition involved executing movements at maximum speed for 10 seconds to assess muscular strength. The second condition required maintaining a constant speed for 30 or 45 seconds using a metronome, focusing on muscular endurance.

Results showed consistent trends across exercises, especially in lunges, knee-ups, and squats. In the constant speed condition, some subjects could not sustain the set tempo due to muscle fatigue, highlighting variations in endurance and strength.

All subjects followed a fixed exercise order: vertical jump, hip-curl, kick-back, reverse lunge, squat, and knee-up (Fig 3). For all exercises except vertical jump, hip-curl, and kick-back, a 10-second repetition test was conducted before and after each exercise to ensure measurement reliability (Fig 2). The constant speed condition was performed for 30 seconds at a predefined tempo: hip-curl (108 bpm), kick-back (108 bpm), reverse lunge (48 bpm), knee-up (132 bpm), and squat (60 bpm).

The key difference between Day 2 and Day 3 was resistance intensity, ranging from 1R to 5R. On Day 2, subjects performed exercises at minimum resistance to familiarize themselves with Bot Fit and proper posture, reducing the risk of injury before the higher resistance exercises on Day 3. Participants in their 20s and 30s reported no significant differences in posture between using minimum resistance and not wearing Bot Fit.

As shown in Fig 4A, at higher resistance levels, subjects struggled to maintain the set speed, indicating muscle fatigue, defined as the point where muscular strength is depleted, preventing further movement. This suggests that high resistance (5R) significantly impacts exercise performance.

Additionally, as observed in Fig 4B and C, when testing different speeds, a threshold speed was identified where some subjects failed to keep pace due to fatigue. Based on this, a high-speed constant tempo was applied to sufficiently challenge muscular endurance.

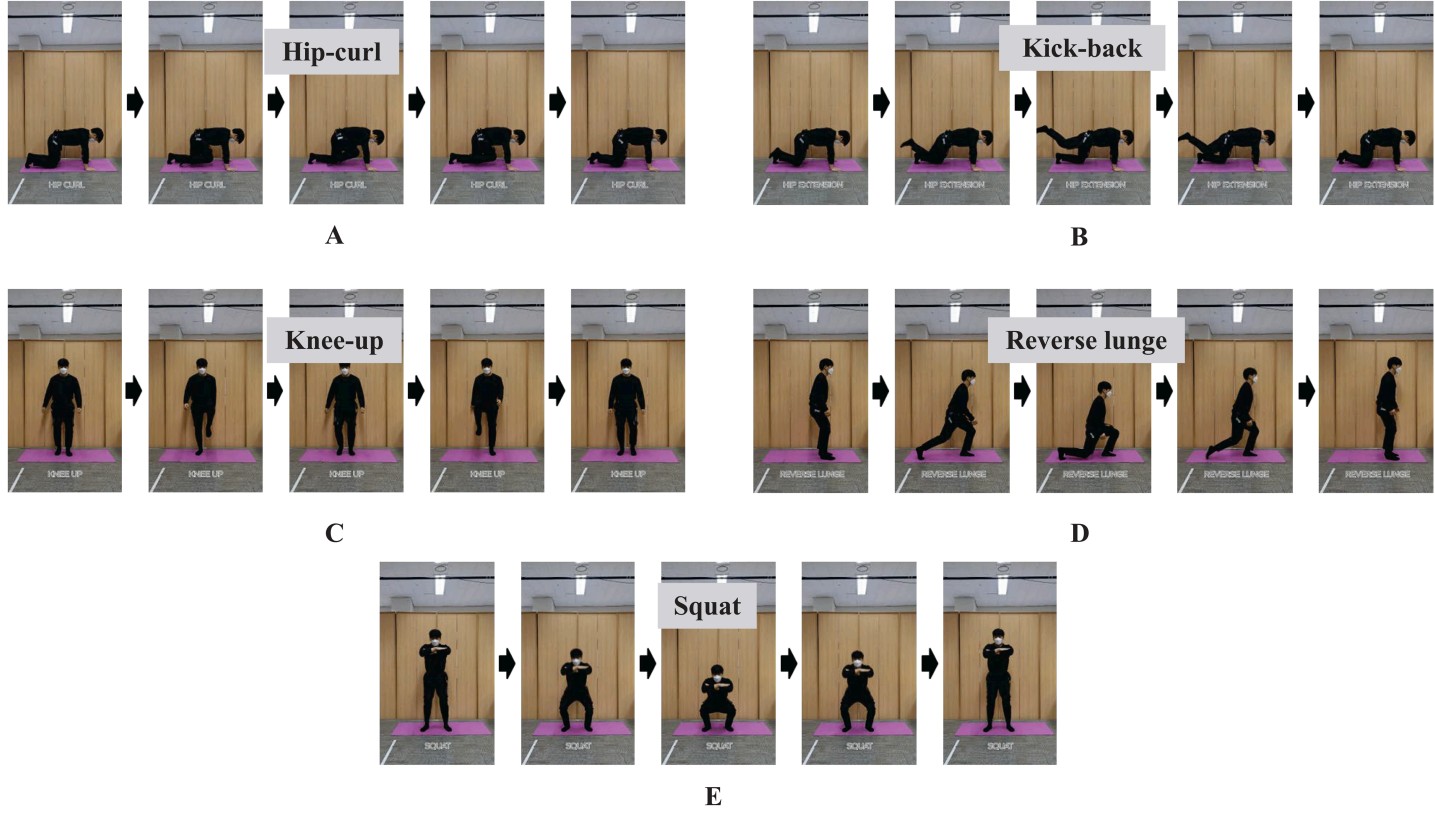

**Fig 3**. **Various resistance exercises performed with Bot Fit during the exercise measurement phase.** (A) Hip-curl. (B) Kick-back. (C) Reverse lunge. (D) Knee-up. (E) Squat.

On Day 3, all exercises were performed at the maximum resistance level (5R), and the study results are based on this session. Exoskeleton robots like Bot Fit provide a controlled and reproducible resistance training environment, offering precise monitoring of muscular strength and new insights into effective strength assessment.

## Data acquisition and processing

**Data acquisition.** Three main types of data were collected in the experiment. The first was baseline muscular strength, measured during the pre-measurement phase. This included muscle parameters derived from body information such as weight, height, BMI, and results from muscular power tests and strength assessments using fitness machines (Table 2).

The second type was motor signals recorded by Bot Fit during the exercise measurement phase. These signals included joint torque (Fig 1, Motor torque) and joint angle, which were used to determine exercise repetitions and speed.

The third type was sEMG signals, collected from wireless sEMG sensors worn alongside Bot Fit. These sensors, attached directly to target muscles, captured muscle activation data, processed as MVIC (Maximum Voluntary Isometric Contraction) [37] (Fig 1, sEMG MVIC). Like motor signals, sEMG data was used to calculate repetitions and exercise speed.

To verify the accuracy of Bot Fit's motor signals, sEMG signals were analyzed since they directly reflect muscle activity. As shown in Fig 4D, a comparison of average exercise speed per repetition between sEMG and motor signals across

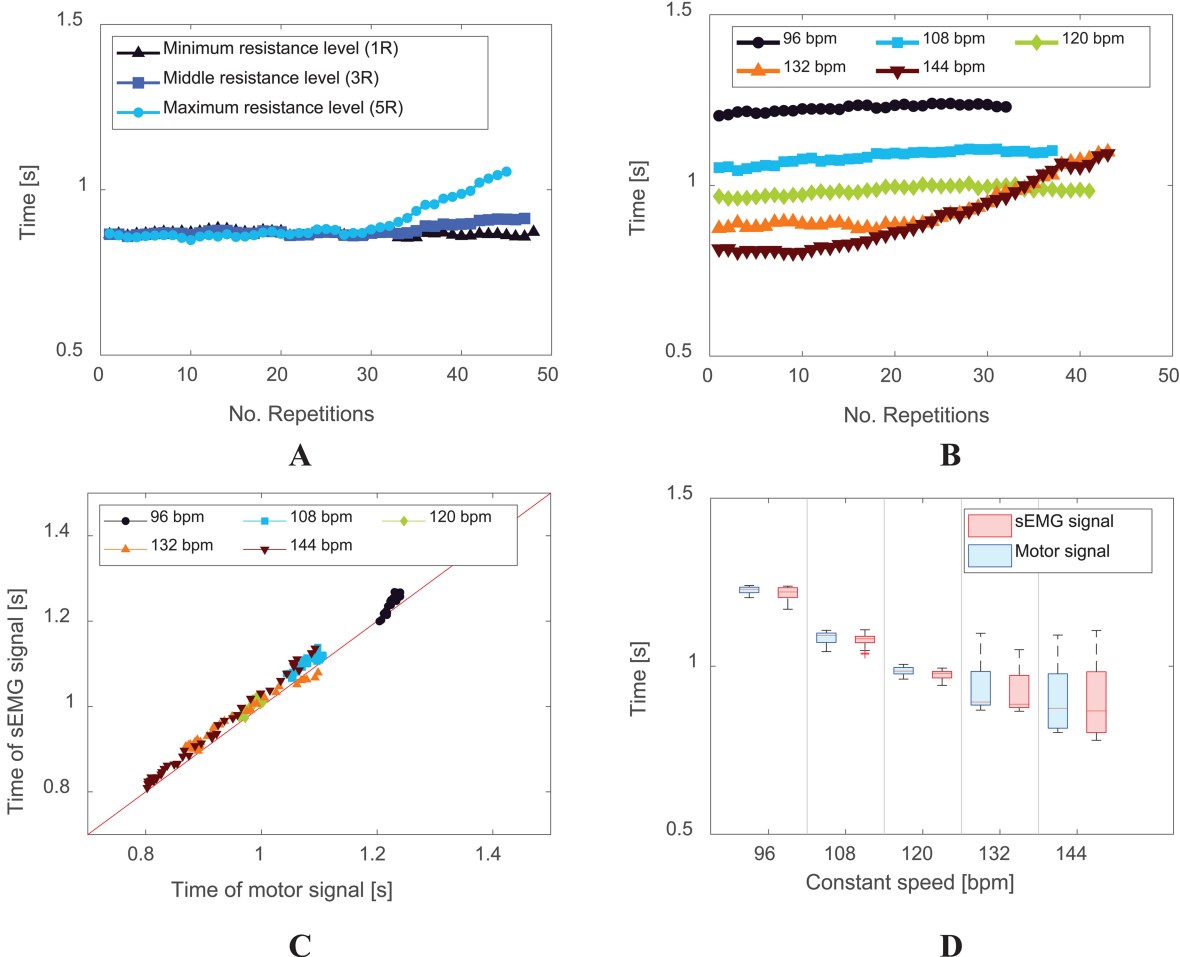

**Fig 4**. **Time intervals in Bot Fit resistance modes, comparing motor signals and sEMG to reflect exercise speed.** (A) Time interval values measured for various resistance levels of Bot Fit: minimum resistance (1R), middle resistance (3R), and maximum resistance (5R). (B) Time intervals of the motor signal analysed at different constant exercise speeds (96 bpm, 108 bpm, 120 bpm, 132 bpm, 144 bpm) using Bot Fit. (C) Similarity comparison between the time intervals based on sEMG signal and motor signal. (D) Box plot results depicting the relationship between constant exercise speed and motor signal (joint torque) derived from Bot Fit, along with the sEMG signal.

different speeds demonstrated their alignment. Additionally, the repetitions recorded by researchers matched those measured by both sEMG and motor signals, confirming their synchronization and ensuring the reliability of motor data in assessing exercise performance.

**Muscle parameters.** We derived muscle parameters by analyzing records obtained from the physical information and exercise equipment of subjects on Day 1 of the experimental protocols to assess the base muscular strength of all subjects (Table 2, S1 File and S2 File).

These parameters were defined using Vertical Jump (VJ) to measure muscle power, and Repetition Maximum (RM) and Isometric contraction test (ISO) through exercise equipment to gauge muscular strength. The strength represented by RM (LE+LC, Weight) and ISO (LE+LC, Weight) was calculated by aggregating the results of leg extension (LE) and leg curl (LC). To facilitate an unbiased comparison among subjects, the results were normalized to each subject's body weight [39].

**Performance metrics.** We utilized the peak values of the joint torque signal, derived from Bot Fit's motor signal, to determine the number of repetitions in each resistance exercise. By analyzing the variations in peak values across repetitions, we calculated the time interval between repetitions, enabling the automatic computation of both repetition count and exercise speed. This process is illustrated in Fig 1, which depicts peak values for each repetition. Similar peak values were also derived from sEMG signals, allowing for a consistent performance evaluation across all resistance exercises during the exercise measurement phase (Table 3).

As shown in Fig 4D, the similarity between the time intervals derived from motor signals and sEMG signals exceeds 0.8, indicating strong temporal alignment. Building on this result, we developed performance metrics based on motor signal analysis for all exercise outcomes. Using the data provided in S3 File, the number of exercise repetitions can be accurately extracted by leveraging various internal motor signals from the robot. These metrics specifically quantify the number of repetitions performed and were designed for Day 3 of the experimental protocol. Data from all exercises conducted on Day 3—including both short-duration and constant-speed protocols—were used to establish these metrics.

As indicated in Table 3, HipCurl Max, KickBack Max, Lunge Max, Kneeup Max, and Squat Max represent the maximum repetition count for each resistance exercise performed on Bot Fit without speed restrictions over 10 seconds. In contrast, HipCurl 108bpm, KickBack 108bpm, Lunge 48bpm, Kneeup 132bpm, and Squat 60bpm refer to repetition counts at a fixed speed (bpm) for an extended duration. While Kneeup is performed for 45 seconds, all other exercises last 30 seconds.

Additionally, we introduced the Constant Speed Zone (CZ), defined as the number of repetitions completed until fatigue in constant-speed exercises over an extended period. This metric evaluates muscular endurance by assessing the ability to maintain exercise speed over time. Using this metric, we developed HipCurl CZ, KickBack CZ, Lunge CZ, Kneeup CZ, and Squat CZ for each exercise.

**Table 3**. **Description of performance metrics.** These metrics, obtained and analysed on Day 3 of the exercise measurement phase, indicate the assessment of resistance exercise performance.

| Terms of metrics | Description |
| --- | --- |
| HipCurl Max | Number of repetitions in the 10s repetition test with no speed limits of hip-curl with the maximum resistance mode of Bot Fit (5R) |
| HipCurl 108bpm | Total number of repetitions in the 30s repetition test of Hip Curl at a constant speed (108bpm) with the maximum resistance mode of Bot Fit (5R) |
| KickBack Max | Number of repetitions in the 10s repetition test with no speed limits of kick-back with the maximum resistance mode of Bot Fit (5R) |
| KickBack 108bpm | Total number of repetitions in the 30s repetition test of Kick Back at a constant speed (108bpm) with the maximum resistance mode of Bot Fit (5R) |
| Lunge Max | Number of repetitions in the 10s repetition test with no speed limits of reverse Lunge with the maximum resistance mode of Bot Fit (5R) |
| Lunge 48bpm | Total number of repetitions in the 30s repetition test of Reverse Lunge at a constant speed (48bpm) with the maximum resistance mode of Bot Fit (5R) |
| Kneeup Max | Number of repetitions in the 10s repetition test with no speed limits of knee-up with the maximum resistance mode of Bot Fit (5R) |
| Kneeup 132bpm | Total number of repetitions in the 45s repetition test of Knee-up at a constant speed (132bpm) with the maximum resistance mode of Bot Fit (5R) |
| Squat Max | Number of repetitions in  10s repetition test with no speed limits of squat with the maximum resistance mode of Bot Fit (5R) |
| Squat 60bpm | Total number of repetitions in the 30s repetition test of Squat at a constant speed (60bpm) with the maximum resistance mode of Bot Fit (5R) |
| HipCurl CZ | Number of repetitions at a constant speed zone (CZ) for 30s in hip-curl |
| KickBack CZ | Number of repetitions at a constant speed zone (CZ) for 30s in kick-back |
| Lunge CZ | Number of repetitions at a constant speed zone (CZ) for 30s in reverse lunge |
| Kneeup CZ | Number of repetitions at a constant speed zone (CZ) for 45s in knee-up |
| Squat CZ | Number of repetitions at a constant speed zone (CZ) for 30s in squat |

We assessed exercise performance using these performance metrics, comparing them with muscle parameters that reflect each subject's baseline muscular strength. This approach provides a comprehensive evaluation of muscular performance while wearing Bot Fit.

## Statistical analysis

We conducted a comprehensive statistical analysis to examine the relationships between muscle parameters and performance metrics in resistance exercises using Bot Fit. The Shapiro-Wilk test was used to assess the normality of muscle parameter data, with p-values above 0.05 indicating a normal distribution (Table 3). To evaluate the strength of associations between muscle parameters and performance metrics, we applied Pearson's correlation coefficient, categorizing correlations as negligible (<0.1), weak (0.1–0.39), moderate (0.4–0.69), strong (0.7–0.79), and very strong (>0.8) [40]. This analysis identified key exercise types and performance metrics that were strongly associated with muscle parameters.

Building on the correlation analysis, we applied Hierarchical Cluster Analysis (HCA), an unsupervised statistical method [41], to classify subjects into three groups based on the selected exercises and performance metrics. The clustering results were compared with muscle parameter data to detect grouping patterns. To evaluate clustering effectiveness, muscle parameter-based clusters were set as the Target class, while performance metric-based clusters were treated as the Predicted class, with precision and recall calculated for each group.

Furthermore, a multivariable linear regression model was developed to estimate muscular strength based on key performance metrics [42,43], offering a practical tool for strength prediction. The model was constructed using ANOVA, with muscle parameters as the dependent variable and performance metrics as independent variables, considering F-values between 0.05 and 0.1. All statistical analyses were performed using SPSS version 26.0 (IBM, Armonk, NY) and MATLAB R2020a (MathWorks, Natick, MA, USA).

## Results

### Correlation with muscle parameters and performance metrics

We investigated the correlation between muscle parameters (Table 2)—VJ, RM (LE+LC, Weight), ISO (LE+LC, Weight) and Total performance—and each performance metric obtained and analysed in the exercise measurement (Table 3).

**Correlation with VJ.**  In Fig 5A, squat exercises show the highest correlation with performance metrics representing vertical jump (VJ) across all subjects.

Notably, the Squat Max metric demonstrates a strong correlation both overall and within specific subgroups, underscoring its effectiveness in assessing the range at which individuals can rapidly maximize their strength and achieve peak speed. Additionally, Squat at 60 bpm also shows significant correlations. Given that the squat is one of the most physically demanding exercises in the protocol, requiring extensive movement, it serves as an excellent indicator of VJ performance. Furthermore, it was observed that the Knee-up at 132 bpm, which evaluates the ability to perform quickly over a relatively long duration, also exhibits a high correlation with performance metrics.

**Correlation with RM (LE+LC, weight).**  In Fig 5B, performance metrics accurately reflecting the muscle parameter RM (LE+LC, Weight) show notable correlations with both squat and lunge exercises.

Squat Max demonstrates a strong correlation with RM (LE+LC, Weight). Additionally, Lunge 48bpm, representing the number of repetitions in a constant speed during exercise, emerges as the second most correlated metric. Squat Max is particularly suitable for evaluating RM (LE+LC, Weight) as it assesses the ability to utilize maximum strength.

**Correlation with ISO (LE+LC, weight).**  In Fig 5C, it is evident that performance metrics associated with lunges, knee-ups, and squats exhibit high correlation values. Notably, Lunge Max, Knee-up at 132 bpm, and Squat Max consistently show strong correlations. This suggests that Squat Max and Lunge Max, which reflect the ability to perform demanding

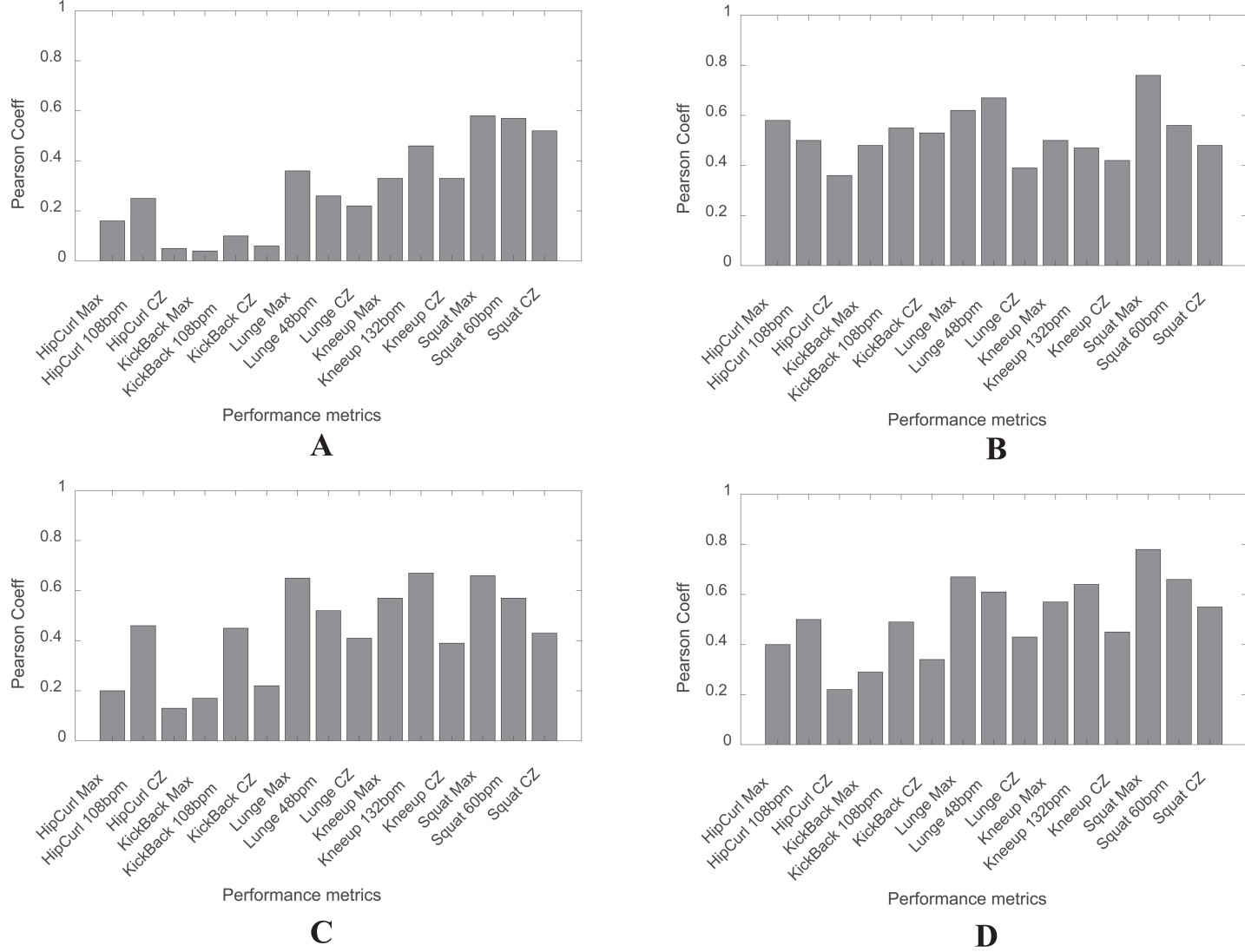

**Fig 5.** **Correlation results between muscle parameters and performance metrics.** (A) Correlation for VJ. (B) Correlation for RM (LE+LC, Weight). (C) Correlation for ISO (LE+LC, Weight). (D) Correlation for Total performance.

exercises in a short period, and Knee-up at 132 bpm, which measures the ability to perform a relatively easier exercise quickly over a longer duration, share a common influence on this muscle parameter.

**Correlation with total performance.** In Fig 5D, which considers VJ, RM (LE+LC, Weight), and ISO (LE+LC, Weight) as overall indicators of muscular strength, Total Performance shows that squat-related metrics have the highest correlations. Additionally, metrics related to lunges and knee-ups also exhibit significant correlations. Notably, Squat Max and Squat at 60 bpm maintain strong correlations, and Lunge Max and Knee-up at 132 bpm also emerge as key metrics representing this muscle parameter.

This indicates that the protocols most strongly correlated with muscular strength are squat, lunge, and knee-up. These protocols are more effective in representing overall muscular strength compared to others, such as hip curl and kick-back.

However, despite being the easiest exercise to perform within this protocol, the Knee-up exercise presents a critical factor related to the angle of exercise (Fig 3D). Due to the very fast pace of the exercise (132 bpm), there is a high likelihood that subjects may struggle to maintain the correct angle compared to other exercise protocols.

Fig 6 illustrates the impact of the exercise angle on performance. Fig 6A shows the results of the hip-joint angles measurable by Bot Fit, observed by the researcher from the subject's side before and after providing correction guidance. According to the study, an angle of 80 degrees or more indicates correct exercise performance, while an angle of 65 degrees or less is not considered accurate. Performing the exercise with a smaller angle makes it relatively easier to perform more repetitions, but performing it at the correct angle allows for a more accurate assessment of the subject's strength as provided in S4 File.

As shown in Fig 6B, the results after angle correction show a high correlation with Total Performance and all Knee-up-related metrics. This confirms that even for the fast-paced Knee-up exercise, careful consideration of the exercise angle is essential.

## HCA performance

As shown in Table 4, we performed HCA on a dataset of 25 subjects, organizing the data into clusters based on muscle parameters and their corresponding performance metrics. Each cluster was subdivided into three groups—high, mid, and low—using performance metrics highly correlated with each muscle parameter. To ensure unbiased clustering, we selected three highly correlated performance metrics per muscle parameter, chosen for their strong correlations across the 25 subjects.

The clustering was conducted for three primary muscle parameters: Vertical Jump (VJ), RM (LE+LC, Weight), and Isometric Contraction (ISO, LE+LC, Weight). For VJ, the performance metrics Squat Max, Squat 60 bpm, and Squat CZ showed strong correlations and were used for clustering. Similarly, Squat Max, Lunge Max, and Lunge 48 bpm correlated strongly with RM (LE+LC, Weight), while Squat Max, Lunge Max, and Knee-up 132 bpm were highly correlated with ISO (LE+LC, Weight).

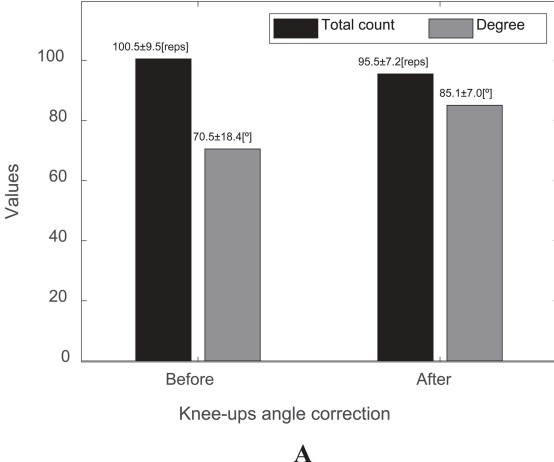

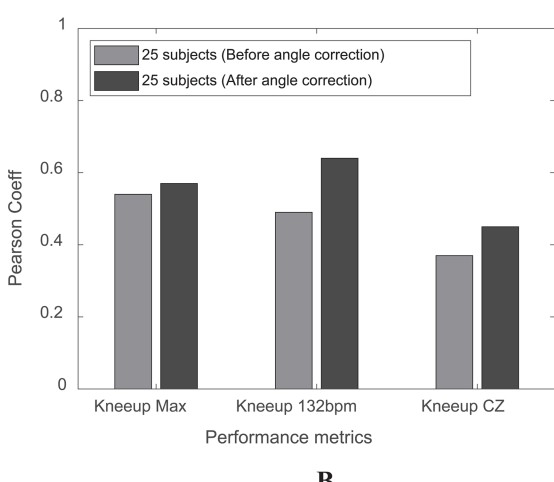

**A**  **B**

**Fig 6. Results of knee-up exercise angle correction.** (A) Comparison of the number of repetitions and exercise angles before and after angle correction during a 45-second knee-up protocol at 132 bpm. The black bars indicate the number of repetitions, while the gray bars denote the exercise angles. (B) Comparison of the correlation between knee-up performance metrics and the muscle parameter, Total Performance, before and after angle correction. The gray bars show the results before angle correction, whereas the black bars depict the correlation results after angle correction.

**Table 4. HCA performance based on muscle parameters (Target) and highly correlated performance metrics (Predicted).** HCA performance is presented for each cluster (High/Mid/Low) in terms of precision, recall, the average of precision and recall, and overall accuracy.

| Muscle parameters (Target) | Performance metrics (Predicted) | HCA performance (Confusion matrix performance) | | | | |
|---|---|---|---|---|---|---|
| | | High | Mid | Low | Avg | Accuracy |
| VJ | Squat Max Squat 60bpm Squat CZ | 0.7 (0.83) | 1.0 (0.83) | 0.6 (1.0) | 0.76 (0.88) | 0.86 |
| RM (LE+LC, Weight) | Squat Max Lunge Max Lunge 48bpm | 0.7 (1.0) | 0.67 (0.83) | 1.0 (0.87) | 0.79 (0.9) | 0.83 |
| ISO (LE+LC, Weight) | Squat Max Lunge Max Kneeup 132bpm | 1.0 (0.8) | 0.9 (1.0) | 0.86 (0.8) | 0.92 (0.86) | 0.94 |
| Total performance | Squat Max Squat 60bpm Lunge Max Lunge 48bpm Kneeup 132bpm | 0.91 (1.0) | 0.85 (1.0) | 1.0 (0.87) | 0.92 (0.95) | 0.89 |

These results confirm that the squat protocol consistently has the highest correlation. Additionally, the lunge and knee-up protocols also exhibit high correlations, effectively representing muscle strength.

## Regression models

Based on the previous correlation and clustering analyses, we developed regression models using performance metrics to estimate the target muscle parameters for all subjects, as shown in Table 5. This approach produced statistically

**Table 5. Performance of regression models for each muscle parameter and associated performance metrics.** Standard errors (SE) reflect the precision of the estimated coefficients, while standardized coefficients (Beta) represent the change in the dependent variable's standard deviations per one-standard-deviation change in the predictor variable [42]. Cohen's $f^2$ [44] quantifies the effect size in regression, indicating the variance explained by an independent variable. Statistical power [45] denotes the likelihood of correctly rejecting the null hypothesis, with a value of 1.0 indicating sufficient sample size and effect size for detecting significant results.

| Model Target | Independent variable | Unstandarised Coeff. | | R (adj. $R^2$) | Effect size; Cohen's $f^2$ (Statistical power) |
|---|---|---|---|---|---|
| | | B | SE | | |
| VJ | (constant) Squat Max | −1.07 1.52 | 12.04 0.435 | 0.589**(0.319) | Large; $f^2 = 0.53$ Power: 0.34 |
| RM (LE+LC, Weight) | (constant) Squat Max | 0.221 0.198 | 0.314 0.034 | 0.775***(0.584) | Large; $f^2 = 1.51$ Power: 0.99 |
| ISO (LE+LC, Weight) | (constant) Squat Max Lunge Max Kneeup 132bpm | −1.06 0.032 0.027 0.013 | 0.442 0.027 0.037 0.005 | 0.763***(0.544) | Large; $f^2 = 1.39$ Power: 0.99 |
| Total perofrmance | (constant) Squat Max Squat 60bpm Lunge Max Kneeup 132bpm | −5.79 0.231 0.061 0.048 0.025 | 1.264 0.078 0.032 0.105 0.016 | 0.861***(0.691) | Large; $f^2 = 2.88$ Power: 1 |

Note: Beta coefficients (B) signify the estimated change in the dependent variable for a one-unit change in the predictor variable, holding all other variables constant. Standard errors (SE) gauge the precision of the estimated coefficients. * $p < 0.05$; ** $p < 0.01$; *** $p < 0.001$. $f^2$: < 0.02 (small effect), $f^2$: 0.02–0.15 (medium effect), and $f^2 > 0.35$ (large effect).

significant models with high correlation and explanatory power (R = 0.861). Notably, the model for estimating total performance was the most effective, with the selected metrics primarily related to squat, lunge, and knee-up protocols.

These findings highlight the specific exercise types most strongly associated with each muscle parameter. VJ and RM (LE+LC, Weight) are closely linked to squat-related metrics, showing a strong correlation with them. Similarly, ISO (LE+LC, Weight) is strongly correlated with metrics tied to squat, lunge, and knee-up exercises. Additionally, total performance is highly correlated with these three exercises, indicating that squats, knee-ups, and lunges are key predictors of overall muscular strength.

## Discussion

This study investigates a method for estimating muscular strength in healthy adults through resistance exercises performed with an exoskeleton robot. To achieve this, Samsung Electronics' exoskeleton robot, Bot Fit, was utilized, and 25 participants performed five resistance exercises to estimate their muscle strength. We hypothesized that resistance exercises performed while wearing an exoskeleton robot could provide a simple yet effective method for estimating muscular strength. As commonly reported in strength assessment studies using repetition-based performance metrics for 1RM estimation in resistance exercises [46], our findings confirm that evaluating resistance exercise performance while wearing an exoskeleton robot correlates strongly with the user's muscular strength.

Despite extensive research on strength assessment, no prior studies have directly estimated muscle strength by evaluating exercise performance under various resistance exercises and conditions. While self-assessment of muscular strength is essential for personal health management, accurate quantification remains challenging without specialized fitness equipment [8,9,47]. To address this, our study establishes a simple yet accurate approach to strength estimation using an exoskeleton robot that provides controlled resistance. Two experimental conditions were introduced: a short-duration, free-speed condition and a fixed-speed condition over an extended period. The first condition mimics 1RM estimation by allowing participants to perform n-RM without a fixed speed, enabling them to exert maximum effort [48]. The second condition maintains a fixed speed for 30 to 45 seconds, emphasizing muscular endurance rather than raw strength. This setup requires users to sustain their movement against resistance, prioritizing endurance over peak force output. The fixed-speed condition was designed to promote greater engagement of both strength and endurance, even during rapid, repetitive exercises. Previous studies indicate that exercise speed significantly influences strength development, with faster exercise speeds yielding greater improvements compared to slower movements [49]. Given the strong relationship between muscular endurance and strength, our study indirectly estimates muscular endurance through strength performance metrics [50,51].

These two experimental conditions provide distinct insights into physical performance. The fixed-speed condition controls for exercise speed, minimizing variability among subjects for objective assessment. In contrast, the maximum-speed condition pushes individuals to exert peak effort in a short time, capturing performance variability across individuals. By incorporating both conditions, we enable a comprehensive evaluation of muscular performance, with fixed speed assessing sustainability, efficiency, and fatigue resistance, and maximum speed measuring peak performance capabilities.

Our results reveal significant correlations between specific resistance exercises and muscular strength indicators. Squat exercise metrics demonstrated the highest correlation with Vertical Jump (VJ), as shown in Fig 5A. This trend was also observed in hierarchical clustering analysis (HCA) results (Table 4), reinforcing the effectiveness of squat-based metrics in representing VJ performance. These findings align with prior studies reporting a strong relationship between squat exercises and vertical jump performance [29]. Squat and lunge exercises exhibited the highest correlation with RM (LE+LC, Weight), as shown in Fig 5B. These exercises primarily target the quadriceps [30] and hamstrings [31], reinforcing their role in lower-limb strength assessment. Lunge exercises were strongly correlated with ISO (LE+LC, Weight), as seen in Fig 5C. The ISO metric, combining leg extension and leg curl measurements, reflects overall lower-limb strength

and power [32,33]. Since the lunge engages both anterior and posterior muscle groups, it was selected as a key performance indicator.

Squat, lunge, and knee-up exercises emerged as the most representative exercises based on total performance indicators (Fig 5D, Table 4). Knee-up exercises, particularly at 132 bpm, were selected for their practicality and strong correlation with lower-body strength and weight management protocols [52–54]. These results were further validated through regression modeling, confirming that squat-related metrics were dominant in strength estimation models, while lunge and knee-up metrics significantly improved model accuracy (Table 5). Among the five exercises tested, squat, lunge, and knee-up were the most effective for muscular strength assessment.

Although Hipcurl and Kickback exercises showed correlations with muscle parameters, they were excluded due to practical limitations. Performing these exercises in a prone position with Bot Fit affected posture and reduced reproducibility, making them less viable for strength assessment. Ultimately, we identified Squat Max (for high-intensity, short-duration performance) and Knee-up 132 bpm (for sustained endurance over time) as the key strength assessment metrics.

Despite the promising findings, this study has several limitations. First, it was conducted with 25 participants, which limits the generalizability of the results. Second, the relatively short exercise durations call for further research to validate the approach over longer training periods. Additionally, variability in exercise angles and individual participant conditions necessitates clearer standardization to ensure consistent exercise form and control for daily physical fluctuations.

One critical factor influencing knee-up performance was the exercise angle, as shown in Fig 6. Because the knee-up exercise was performed at a high cadence (132 bpm), a lower knee angle made it easier to maintain speed. To ensure consistency and accurate assessment [55], we monitored movement patterns using Bot Fit's motor signals and direct observation. Participants maintained a knee-up angle of less than 65 degrees, which enabled controlled and reproducible performance.

Beyond its application to strength assessment in healthy adults, the Bot Fit system may also hold promise for clinical and rehabilitation settings. Although this study focused on young, healthy individuals, the wearable robotic platform could potentially be adapted to support individuals with reduced mobility or those undergoing physical therapy. Future studies involving older adults or clinical populations will be necessary to evaluate the system's usability and effectiveness in these contexts.

Nevertheless, this study introduces an innovative approach to muscular strength assessment using the hip-joint exoskeleton robot, Bot Fit. By addressing the limitations of traditional strength assessment methods, our findings underscore the potential of wearable robotics to enable dynamic, real-time, and accessible evaluation of muscular performance across various settings.

## Conclusion

Assessing muscular strength is fundamental to overall well-being and exercise performance. This study introduced Bot Fit, a hip-joint exoskeleton robot, as an innovative tool for evaluating muscular strength in healthy individuals through safe and controlled resistance exercises, with automated performance analysis based on motor signals.

A total of 25 participants assessed their baseline muscular strength and performed five resistance exercises using Bot Fit. The analysis revealed strong correlations between muscular strength and exercise performance, particularly in squats, lunges, and knee-ups. Notably, complex movements such as squats and lunges were associated with short-duration, high-intensity performance, whereas simpler exercises like knee-ups reflected sustained performance at high speeds. These findings indicate that Bot Fit offers a precise, reproducible, and convenient method for strength assessment, eliminating the need for additional equipment or human supervision.

Beyond general fitness applications, Bot Fit may also hold potential in clinical and rehabilitation settings, particularly for individuals with limited mobility. With further validation and refinement, the system could support broader use across different populations and health conditions.

Future research should expand the sample size and incorporate additional data sources, such as sEMG and motor signal analysis, to improve model validity and robustness. Through continued development, Bot Fit may serve as a valuable platform for both fitness evaluation and rehabilitation support, contributing to more efficient and accessible muscular strength assessment.

## Acknowledgments

We would like to thank all the participants who voluntarily took part in this study.

## Supporting information

**S1 File.** `muscle_param.csv`. Anonymized muscle strength parameters for 25 participants, used in the regression analyses reported in the manuscript. All personally identifiable information has been removed in accordance with IRB-approved protocol.
(CSV)

**S2 File.** `demographics.csv`. Anonymized demographic information (age, height, weight, BMI) for 25 participants, corresponding to the summary statistics reported in Table 1.
(CSV)

**S3 File.** `angle_data.zip`. Raw joint angle data (left and right legs) collected from 25 participants across all exercise protocols. This compressed folder contains individual participant files (`sub1−sub25`) with time-series signals representing lower-limb motion recorded by the Bot Fit system during resistance exercise trials.
(ZIP)

**S4 File.** `kneeup_correction.csv`. Processed knee-up exercise data used in the analysis presented in Fig 6, including raw and corrected repetitions and angle measurements derived from the raw joint angle data.
(XLSX)

## Author contributions

**Conceptualization:** Byungmun Kang, DaeEun Kim.

**Data curation:** Byungmun Kang.

**Formal analysis:** Byungmun Kang.

**Funding acquisition:** Byungmun Kang.

**Investigation:** Byungmun Kang.

**Methodology:** Byungmun Kang.

**Resources:** Byungmun Kang.

**Software:** Byungmun Kang.

**Supervision:** Byungmun Kang.

**Validation:** Byungmun Kang.

**Visualization:** Byungmun Kang.

**Writing – original draft:** Byungmun Kang.

**Writing – review & editing:** Byungmun Kang, DaeEun Kim, Changmin Lee, Dongwoo Kim, Hwang-Jae Lee, Dokwan Lee, YoonMyung Kim, Hyung-Gyu Jeon, Kyunghwan Jung.

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
