## [Decision Letter · Decision Letter 0]

29 May 2025

PONE-D-25-09036Bot Fit: A Novel Approach to Assessing Lower Limb Muscular StrengthPLOS ONE

Dear Dr. Kim,

Thank you for submitting your manuscript to PLOS ONE. After careful consideration, we feel that it has merit but does not fully meet PLOS ONE’s publication criteria as it currently stands. Therefore, we invite you to submit a revised version of the manuscript that addresses the points raised during the review process.

We look forward to receiving your revised manuscript.

Kind regards,

Hasan Sozen

Academic Editor

PLOS ONE

Journal Requirements:

“This study was supported by Samsung Electronics, Republic of Korea. This work was supported by the National Research Foundation of Korea (NRF) grant funded by the Korea government (MSIT) (Grant No. NRF-2020S1A5A2A03044672, 2020R1A2B5B01002395).”

“This study was supported by Samsung Electronics, Republic of Korea. This work was supported by the National Research Foundation of Korea (NRF) grant funded by the Korea government (MSIT) (Grant No. NRF-2020S1A5A2A03044672, 2020R1A2B5B01002395).”

“The author(s) received no specific funding for this work”

5. In the online submission form, you indicated that “The datasets used and analyzed in the manuscript are available from the corresponding author upon reasonable request (Email:

kbmang@yonsei.ac.kr, daeeun@yonsei.ac.kr). The dataset includes personal information such as age, height, weight, and blood pressure; therefore, access to the data is restricted to ensure the privacy and confidentiality of participants.”.

Reviewers' comments:

Reviewer's Responses to Questions

**Comments to the Author**

1. Is the manuscript technically sound, and do the data support the conclusions?

Reviewer #1: Yes

Reviewer #2: Yes

2. Has the statistical analysis been performed appropriately and rigorously?

Reviewer #1: Yes

Reviewer #2: Yes

3. Have the authors made all data underlying the findings in their manuscript fully available?

Reviewer #1: No

Reviewer #2: Yes

4. Is the manuscript presented in an intelligible fashion and written in standard English?

Reviewer #1: Yes

Reviewer #2: Yes

5. Review Comments to the Author

Reviewer #1: Dear Authors,

Thank you for the opportunity to review your manuscript. Your study addresses a timely and relevant topic in the fields of biomechanics and physical performance assessment, offering an innovative perspective on wearable technology for strength evaluation. Overall, your manuscript is well-structured, methodologically sound, and clearly written. Your experimental design is solid, with clear objectives, appropriate controls, and well-defined procedures. The sample size, while modest, is adequate for the exploratory nature of the study, and ethical approval is appropriately addressed. The statistical approach is thorough and well-executed. You employed appropriate tests (e.g., Pearson correlation, regression modeling, HCA) and supported your findings with effect sizes and power analysis, which strengthens the credibility of your results. Including accuracy metrics and classification performance for cluster analysis was a notable strength. The manuscript is written in clear, standard English. The structure flows logically, and the scientific language is appropriate for an academic audience. Only minor stylistic refinements may be necessary at revision for polishing, but these do not hinder readability. You have transparently acknowledged the main limitations (e.g., sample size, variation in technique), and your conclusions are well-supported by the presented data. However, according to PLOS ONE’s policies, data must be fully and publicly available. Currently, the manuscript states that data are available upon request, due to privacy concerns. I suggest that you anonymize the dataset (removing identifiable features) and deposit it in a public repository. Alternatively, provide a formal justification from your IRB or ethics committee explaining the restriction.

Minor Suggestions for Improvement:

Consider including a brief discussion on the potential application of Bot Fit in clinical or rehabilitation contexts. Clarify whether the Bot Fit system could be adapted to populations with mobility impairments, given the emphasis on healthy young adults. In summary, this is a well-conducted study with methodological rigor and practical relevance. With minor revisions — particularly regarding data availability — I believe the manuscript is suitable for publication.

Sincerely,

Rodrigo Oliveira

Reviewer #2: It is a valuable paper, edited in a clear and easy-to-follow style. Iconografy facilitates the understanding of the paper. The results are valuable, as it proves the efficacy of a innovative tool in evaluating muscular strength, which might be useful for rehabilitation.

6. PLOS authors have the option to publish the peer review history of their article (what does this mean?). If published, this will include your full peer review and any attached files.

Reviewer #1: **Yes:** Rodrigo Assunção de Oliveira

Reviewer #2: **Yes:** Diana Ciubotariu

---

## [Author Response · Author response to Decision Letter 1]

11 Nov 2025

Dear Editor and Reviewers:

Our previous paper, [PONE-D-25-09036] - [EMID:834ef8ac76b1d107], entitled " Bot Fit: A Novel Approach to Assessing Lower Limb Muscular Strength," has been revised in response to your valuable comments. We appreciate your insightful feedback and have made the following modifications accordingly.

We would like to express our sincere gratitude once again for your invaluable comments and suggestions.

Sincerely,

DaeEun Kim

Biological Cybernetics Lab

School of Electrical & Electronic Engineering

Yonsei University

Sinchon, Seodaemoon-gu

Seoul, 120-749

South Korea

Journal Requirements:

We thank the editor for the guidance regarding PLOS ONE’s formatting requirements. As suggested, we have carefully revised the manuscript to ensure it aligns with the journal’s style guidelines. Specifically, we corrected formatting elements such as removing periods after figure and table labels (e.g., changing “Fig.” to “Fig” and ensuring consistency in table references). We have also reviewed file naming conventions and updated all supplementary materials accordingly.

Please let us know if any additional formatting adjustments are required.

Thank you for pointing out the inconsistency between the ‘Funding Information’ and ‘Financial Disclosure’ sections. As suggested, we have carefully reviewed and revised both sections to ensure that the grant numbers and funding sources are fully aligned and accurately reported. The corrected and consistent grant information is now reflected in the resubmitted manuscript.

“This study was supported by Samsung Electronics, Republic of Korea. This work was supported by the National Research Foundation of Korea (NRF) grant funded by the Korea government (MSIT) (Grant No. NRF-2020S1A5A2A03044672, 2020R1A2B5B01002395).”

Thank you for your guidance regarding the funding disclosure. As requested, we have added a statement clarifying the role of the funders both in the cover letter and in the manuscript.

Cover Letter

Funding Statement:

This study was supported by Samsung Electronics, Republic of Korea. This work was also funded by the National Research Foundation of Korea (NRF) through a grant from the Korean government (MSIT) (Grant No. 2020R1A2B5B01002395), and by the Korea Institute for Advancement of Technology (KIAT) through a grant from the Korean government (MOTIE) (Grant No. P0020535, The Competency Development Program for Industry Specialists).

Role of Funders:

Manuscript – Funding Statement Section:

The authors declare that financial support was received for the research, authorship, and/or publication of this article. This study was supported by Samsung Electronics, Republic of Korea. Additional support was provided by the National Research Foundation of Korea (NRF), funded by the Korean government (MSIT) (Grant No. 2020R1A2B5B01002395), and by the Korea Institute for Advancement of Technology (KIAT), funded by the Korean government (MOTIE) (Grant No. P0020535, The Competency Development Program for Industry Specialists).

“This study was supported by Samsung Electronics, Republic of Korea. This work was supported by the National Research Foundation of Korea (NRF) grant funded by the Korea government (MSIT) (Grant No. NRF-2020S1A5A2A03044672, 2020R1A2B5B01002395).”

“The author(s) received no specific funding for this work”

Thank you for pointing this out. In response to your comment, we have removed all funding-related information from the Acknowledgments section and other parts of the manuscript. Instead, we have incorporated the appropriate funding information exclusively into the Funding Statement section.

The revised Funding Statement now reads as follows:

Funding Statement

The author(s) declare that financial support was received for the research, authorship, and/or publication of this article. This study was supported by Samsung Electronics, Republic of Korea. Additional support was provided by the National Research Foundation of Korea (NRF), funded by the Korean government (MSIT) (Grant No. 2020R1A2B5B01002395), and by the Korea Institute for Advancement of Technology (KIAT), funded by the Korean government (MOTIE) (Grant No. P0020535, The Competency Development Program for Industry Specialists).

We confirm that this revised Funding Statement is included in both the cover letter and the manuscript, and that all other mentions of funding have been removed as per the journal’s guidelines.

5. In the online submission form, you indicated that “The datasets used and analyzed in the manuscript are available from the corresponding author upon reasonable request (Email:

kbmang@yonsei.ac.kr, daeeun@yonsei.ac.kr). The dataset includes personal information such as age, height, weight, and blood pressure; therefore, access to the data is restricted to ensure the privacy and confidentiality of participants.”.

We thank the editors for highlighting the importance of data availability. In response, we have revised our data availability plan to fully comply with PLOS ONE’s open data policies while maintaining adherence to ethical and contractual obligations.

We now provide four anonymized datasets as Supporting Information:

• muscle_param.csv: Contains muscle strength parameters used in our regression analysis, with all personally identifiable information (e.g., age, height, weight, blood pressure) removed in accordance with Section 10 of our IRB-approved protocol.

• demographics.csv: Includes participants’ general demographic information (age, height, weight, and BMI) without any identifiers.

• angle_data.zip: Includes raw joint angle data (left and right legs) collected from 25 participants (sub1 to sub25) during each exercise protocol. These data reflect movement characteristics only and contain no sensitive or identifiable information.

• kneeup_correction.csv: Contains processed knee-up exercise data used to generate Figure 6 in the manuscript.

However, we are unable to share the full raw dataset generated by the Bot Fit system, which includes proprietary motor signals, internal performance computations, and signal processing algorithms, due to confidentiality agreements with Samsung Electronics. The Bot Fit system remains under active development, and these data contain commercially sensitive information.

We are prepared to provide supporting documentation, including email correspondence with Samsung Electronics, to verify these restrictions. Therefore, we respectfully request an exemption from full data disclosure based on these ethical and contractual constraints.

We believe that the anonymized datasets provided as Supporting Information offer sufficient transparency and reproducibility for the scientific community.

Additionally, the Data Availability Statement in the manuscript has been revised as follows:

Data Availability Statement

Anonymized datasets sufficient to replicate the analyses and findings reported in the manuscript are provided as Supporting Information:

(1) muscle_param.csv, containing muscle strength parameters used in the regression analysis;

(2) demographics.csv, containing participant age, height, weight, and BMI;

(3) angle_data.zip, including raw joint angle data (left and right legs) from all exercise protocols; and

(4) kneeup_correction.csv, containing processed knee-up exercise data underlying Figure 6.

Researchers with justified purposes may contact the corresponding author for general inquiries about the study.

However, all data requests must be directed to a non-author data contact:

Minchul Kim, Samsung Electronics Co., Ltd.

Email: min1chul.kim@samsung.com

This contact will review and handle data requests in accordance with Samsung Electronics’ confidentiality and intellectual property policies.

We thank the editors for this helpful instruction regarding Supporting Information. In response, we have added clear captions for the Supporting Information files at the end of the manuscript, in accordance with the journal’s guidelines.

The following files have been included:

S1 Dataset. muscle_param.csv. Anonymized dataset of muscle strength parameters for 25 participants used in the statistical analyses. All personally identifiable information has been removed in accordance with IRB guidelines.

S2 Dataset. demographics.csv. Anonymized demographic information (age, height, weight, BMI) for 25 participants, corresponding to Table1 in the manuscript.

S3 Dataset. angle_data.zip. Compressed folder containing left and right leg joint angle data for each exercise protocol across 25 participants (sub1–sub25). These raw signals represent lower-limb motion captured by the Bot Fit system during resistance exercise trials.

S4 Dataset. kneeup_correction.csv. Processed knee-up exercise data underlying Figure6, including raw and corrected repetitions and angle measurements.

We have also updated the in-text citations throughout the manuscript to appropriately reference these Supporting Information files.

Thank you for your comment. We have carefully reviewed our reference list to ensure its completeness and accuracy. We confirm that none of the references cited in our manuscript have been retracted, and therefore no changes to the reference list were necessary.

If any issues regarding the cited literature arise during the editorial process, we will gladly address them without delay.

Reviewers' comments:

Reviewer's Responses to Questions

Comments to the Author

1. Is the manuscript technically sound, and do the data support the conclusions?

Reviewer #1: Yes

Reviewer #2: Yes

We sincerely thank Reviewer #1 and Reviewer #2 for their positive evaluations regarding the technical soundness of our study and the strength of the evidence supporting our conclusions. We appreciate your recognition of the rigor in our experimental design, including appropriate controls, replication, and sample size. Your feedback reinforces our confidence in the validity and relevance of our findings.________________________________________

2. Has the statistical analysis been performed appropriately and rigorously?

Reviewer #1: Yes

Reviewer #2: Yes

We sincerely thank Reviewer #1 and Reviewer #2 for recognizing the rigor and appropriateness of our statistical analyses. We carefully selected methods such as correlation analysis, regression modeling, and cluster validation in accordance with our study objectives and data characteristics. Your acknowledgment reinforces the reliability and validity of our analytical approach.________________________________________

3. Have the authors made all data underlying the findings in their manuscript fully available?

Reviewer #1: No

Reviewer #2: Yes

We thank the reviewers for highlighting the importance of data availability. In response to Reviewer #1’s concern, we have revised our data availability plan to fully comply with the PLOS ONE Data Availability policy while maintaining adherence to ethical and contractual obligations.

We now provide four anonymized datasets as Supporting Information:

• muscle_param.csv: Contains muscle strength parameters used in the regression analysis, with all personally identifiable information (e.g., age, height, weight, blood pressure) removed in accordance with Section 10 of our IRB-approved protocol.

• demographics.csv: Includes participants’ demographic data (age, height, weight, and BMI) in de-identified form.

• angle_data.zip: Contains raw joint angle data (left and right legs) from all exercise protocols, reflecting movement characteristics only and excluding any sensitive or personally identifiable

---

## [Decision Letter · Decision Letter 1]

3 Dec 2025

Bot Fit: A Novel Approach to Assessing Lower Limb Muscular Strength

PONE-D-25-09036R1

Dear Dr. Kim,

We’re pleased to inform you that your manuscript has been judged scientifically suitable for publication and will be formally accepted for publication once it meets all outstanding technical requirements.

Kind regards,

Hasan Sozen

Academic Editor

PLOS ONE

Reviewers' comments:

Reviewer's Responses to Questions

**Comments to the Author**

1. If the authors have adequately addressed your comments raised in a previous round of review and you feel that this manuscript is now acceptable for publication, you may indicate that here to bypass the “Comments to the Author” section, enter your conflict of interest statement in the “Confidential to Editor” section, and submit your "Accept" recommendation.

Reviewer #1: All comments have been addressed

2. Is the manuscript technically sound, and do the data support the conclusions?

Reviewer #1: Yes

3. Has the statistical analysis been performed appropriately and rigorously?

Reviewer #1: Yes

4. Have the authors made all data underlying the findings in their manuscript fully available?

Reviewer #1: Yes

5. Is the manuscript presented in an intelligible fashion and written in standard English?

Reviewer #1: Yes

6. Review Comments to the Author

Reviewer #1: Dear Authors,

Thank you for the careful and comprehensive revision of your manuscript. I appreciate the thorough point-by-point rebuttal and the tracked-changes manuscript. The revisions have materially improved the manuscript. The revised manuscript satisfactorily addresses the majority of reviewers’ requests. The authors state that full raw motor-signal data and internal processing steps remain proprietary and cannot be publicly released due to contractual restrictions with Samsung Electronics. The authors have provided anonymized datasets (S1–S4) that include the variables necessary to reproduce the reported analyses and have offered to supply documentation (e.g., correspondence with Samsung) and a non-author contact at Samsung to handle data requests. In my assessment, the anonymized SI supplied appears sufficient to reproduce the statistical results presented in the manuscript. However, the authors’ request for an exemption from sharing the proprietary raw signal data is a matter for editorial/legal assessment under PLOS policy. If the Editors require further evidence to grant the exemption, the documentation offered by the authors should be requested and examined.

In short, provided that the editorial office accepts the authors’ approach to data sharing (anonymized SI plus restricted proprietary raw data with documented justification), the manuscript is suitable for publication.

Sincerely,

Dr. Rodrigo Assunção de Oliveira

7. PLOS authors have the option to publish the peer review history of their article (what does this mean?). If published, this will include your full peer review and any attached files.

Reviewer #1: **Yes:** Rodrigo Assunção de Oliveira

---

## [Editor Report · Acceptance letter]

PONE-D-25-09036R1

PLOS One

Dear Dr. Kim,

I'm pleased to inform you that your manuscript has been deemed suitable for publication in PLOS One. Congratulations! Your manuscript is now being handed over to our production team.

Kind regards,

on behalf of

Assoc. Prof. Hasan Sozen

Academic Editor

PLOS One